

# Identification of hub genes and small-molecule compounds related to intracerebral hemorrhage with bioinformatics analysis

Zhendong Liu[1,2,*], Ruotian Zhang[1,2,*], Xin Chen[1,2], Penglei Yao[1,2], Tao Yan[1,2], Wenwu Liu[1,2], Jiawei Yao[1,2], Andrei Sokhatskii[3], Ilgiz Gareev[3] and Shiguang Zhao[1,2]

[1] The First Affiliated Hospital of Harbin Medical University, Department of Neurosurgery, Harbin, Heilongjiang Province, People's Republic of China
[2] Harbin Medical University, Institute of Brain Science, Harbin, Heilongjiang Province, People's Republic of China
[3] Bashkir State Medical University, Ufa, Russia
[*] These authors contributed equally to this work.

Corresponding author
Shiguang Zhao,
guangsz@hotmail.com

## ABSTRACT

**Background**. Because of the complex mechanisms of injury, conventional surgical treatment and early blood pressure control does not significantly reduce mortality or improve patient prognosis in cases of intracerebral hemorrhage (ICH). We aimed to identify the hub genes associated with intracerebral hemorrhage, to act as therapeutic targets, and to identify potential small-molecule compounds for treating ICH.

**Methods**. The GSE24265 dataset, consisting of data from four perihematomal brain tissues and seven contralateral brain tissues, was downloaded from the Gene Expression Omnibus (GEO) database and screened for differentially expressed genes (DEGs) in ICH, with a fold change (FC) value of ($|\log2FC|$) > 2 and a $P$-value of <0.05 set as cut-offs. The functional annotation of DEGs was performed using Gene Ontology (GO) resources, and the cell signaling pathway analysis of DEGs was performed using the Kyoto Encyclopedia of Genes and Genomes (KEGG), with a $P$-value of <0.05 set as the cut-off. We constructed a protein-protein interaction (PPI) network to clarify the interrelationships between the different DEGs and to select the hub genes with significant interactions. Next, the DEGs were analyzed using the CMap tool to identify small-molecule compounds with potential therapeutic effects. Finally, we verified the expression levels of the hub genes by RT-qPCR on the rat ICH model.

**Result**. A total of 59 up-regulated genes and eight down-regulated genes associated with ICH were identified. The biological functions of DEGs associated with ICH are mainly involved in the inflammatory response, chemokine activity, and immune response. The KEGG analysis identified several pathways significantly associated with ICH, including but not limited to HIF-1, TNF, toll-like receptor, cytokine-cytokine receptor interaction, and chemokine molecules. A PPI network consisting of 57 nodes and 373 edges was constructed using STRING, and 10 hub genes were identified with Cytoscape software. These hub genes are closely related to secondary brain injury induced by ICH. RT-qPCR results showed that the expression of ten hub genes was significantly

increased in the rat model of ICH. In addition, a CMap analysis of three small-molecule compounds revealed their therapeutic potential.

**Conclusion**. In this study we obtained ten hub genes, such as IL6, TLR2, CXCL1, TIMP1, PLAUR, SERPINE1, SELE, CCL4, CCL20, and CD163, which play an important role in the pathology of ICH. At the same time, the ten hub genes obtained through PPI network analysis were verified in the rat model of ICH. In addition, we obtained three small molecule compounds that will have therapeutic effects on ICH, including Hecogenin, Lidocaine, and NU-1025.

## INTRODUCTION

On account of its high morbidity and mortality, intracerebral hemorrhage (ICH) is a worldwide public health concern (*Qureshi et al., 2001*). About 2 million people experience ICH every year, and its incidence is still rising (*Labovitz et al., 2005*; *Qureshi et al., 2007*). Half of patients die within a year, and two-third of the survivors become permanently disabled (*Fogelholm et al., 2005*). The poor prognosis of patients with ICH is mainly related to its complex pathogenesis. The pathological process of ICH can be divided into primary injury, which consists of mechanical compression caused by hematoma, and secondary injury, which is caused by various factors such as cytotoxicity of blood, hypermetabolism, excitotoxicity, oxidative stress and inflammation (*Aronowski & Zhao, 2011*). Several studies have shown that treatments targeting the ICH-induced cellular processes and their downstream molecules may be effective (*Morioka & Orito, 2017*).

According to previous reports, the cytokines released by various inflammatory cells such as leukocytes, macrophages, and microglia after the onset of ICH were involved in the formation of secondary brain injury caused by ICH (*Wang & Dore, 2007*). For example, microglia/macrophages activated after the ICH event can cause an increase in the expression levels of TNF-a and IL-1$\beta$, which can aggravate brain damage (*Aronowski & Hall, 2005*; *Wagner, 2007*). In the ICH model constructed by animals, the expression of TNF-$\alpha$ was significantly increased, which promoted the formation of cerebral edema (*Hua et al., 2006*). Toll-like receptors (TLRs) play an irreplaceable role in the regulation of immune responses and inflammatory responses (*Akira & Takeda, 2004*; *Takeda & Akira, 2004*). TLRs are a family of transmembrane proteins, and it has been reported in the literature that the significant increase in expression levels of TLR2 and TLR4 in peripheral monocytes of ICH patients is significantly associated with poor prognosis in patients (*Rodriguez-Yanez et al., 2012*). In addition, the increased expression levels of proinflammatory genes such as CD36, iNOS, and MMP-9 all aggravated the pathological progression of ICH (*Zhao et al., 2009*; *Zhao et al., 2007*). Therefore, we have reason to speculate that there should be many potential genes in the pathological process of ICH that may play important regulatory

roles. However, the expression profile and function of mRNAs in the course of ICH have not yet been fully explained.

Presently, research on ICH treatment has mostly focused on the clinical aspects of therapeutic interventions, and investigations into the molecular mechanisms underlying the pathology of ICH remain scarce (*Yin et al., 2017*). Therefore, identifying the therapeutic targets for ICH at the molecular level may improve the existing treatment strategies for ICH. Although previous reports by *Rosell et al. (2011)* on the expression profile of mRNAs have revealed the differentially expressed genes (DEGs) in the pathology of ICH, the main focus of the study was on the top ten up-regulated genes and ten down-regulated genes with the difference of the largest fold change. However, their research ignores the interaction between DEGs, which is crucial to understanding the biological significance of DEGs.

Our current research goal is to explore the expression profiles of mRNAs in ICH patients to screen for hub genes through the PPI network and subsequently analyze their functions. Furthermore, the rat model of ICH has been established to verify the expression level of hub genes by RT-qPCR. In addition, the DEGs were analyzed using the CMap tool to identify small-molecule compounds with potential therapeutic effects on ICH.

## MATERIALS & METHODS

### Data collection

The gene expression profiles of patients with ICH were obtained from the GEO database (https://www.ncbi.nlm.nih.gov/geo/) (*Clough & Barrett, 2016*), a public database for researchers. GSE24265, which includes the data from four spontaneous cases of intracerebral hemorrhage, relied on the Agilent GPL570 platform (Affymetrix Human Genome U133 Plus 2.0 Array). All brain tissue samples were from four patients with spontaneous intracerebral hemorrhage who were diagnosed within the previous four days, including two men and two women, with a median age of 79 years (68–92 years), and all samples were obtained within 5 h of the patients' death. Perihematomal areas were used for the disease groups, while the contralateral grey and white matter were used as controls (*Rosell et al., 2011*).

### Screening results of DEGs

We downloaded the series matrix file dataset of GSE24265, and subsequently transformed the gene probes of the platform to gene names by referencing the GPL570 platform. Both the normalization of the data and screening of the DEGs were performed using the limma package in the R language (version 3.5.1) (*Guo et al., 2017*). To identify the genes with significant differences in the fold change of expression, we set the screening criteria to filter genes with a fold change (FC) value of ($|\log2FC|$) > 2 and a $P$-value of < 0.05.

### GO and KEGG pathway analyses for identification of DEGs

To enable better recognition of the biological functions of DEGs, we used online tools to perform GO enrichment and KEGG analyses. Based on the description of the GO analysis, the gene function annotations were identified as biological processes (BP), cellular components (CC), or molecular functions (MF). DAVID (https://david.ncifcrf.gov/) and

KOBAS (version 3.0; https://bio.tools/kobas) are considered versatile bioinformatics tools; their main functions include gene annotation, visualization, identification of signaling pathways, and integrated functional discovery. Functional annotation analysis of the DEGs was performed using DAVID, and the KEGG-assisted cell signaling pathway analysis of the DEGs was carried out using KOBAS (*Tang et al., 2018*). *P*-values of $< 0.05$ were considered to be statistically significant.

## Construction of the PPI network and identification of hub genes

STRING is a public database that contains interactions between known and predicted proteins (https://string-db.org/) (*Szklarczyk et al., 2019*). PPI is essential for studying protein function since it can help elucidate the function of regulation among proteins. We uploaded the DEGs obtained from the GSE24265 dataset onto STRING's official website to obtain the interrelationships between proteins and to set the minimum required interaction score to 0.15 to visualize the interaction networks with Cytoscape (version 3.2.1) (*Shannon et al., 2003*). The screening of hub genes was based on the degree of connectivity between DEGs and genes; the DEGs featuring the ten highest degrees were recognized as hub genes.

## Obtaining small-molecule compounds for the treatment of ICH by CMap analysis

CMap is a gene-expression-based drug development system that explains associations among genes, drugs, and diseases by integrating the effects of thousands of small-molecule drugs on various human cells (*Lamb et al., 2006*). In this study, we converted the DEGs into probes and subsequently uploaded them onto the official website of CMap to obtain corresponding small-molecule compounds. The negatively related drugs ($P < 0.01$ and enrichment $< 0$) were considered to be therapeutically effective for treating ICH.

## Rat model of ICH

All animals used in this study were tested according to the Guide for Animal Experimentation of Harbin Medical University, and the animal study approved by the Institutional Animal Care and Use Committee at Harbin Medical University. Six adult male Sprague-Dawley rats, weighing 200–250 g, aged 8 weeks, provided by the Animal Experimental Center, Harbin Medical University, were used in the present study. The animals, fixed using a stereotaxic frame, were intraperitoneally injected with sodium pentobarbital (50 mg/kg), and, using a skull drill (about one mm in diameter), holes were drilled near the right coronal suture, 3.5 mm outside the midline. Autologous whole blood (50 μL) was collected from the tail vein using a sterile disposable syringe, and, using the same syringe, the collected autologous blood was slowly injected into the right basal ganglia at a rate of 10 μl/min (*He et al., 2018*). After ten minutes, the syringe was pulled out and the skin was sutured. Six rats were equally divided into experimental and control groups, but the control group only received a needle insertion. All rats were intraperitoneally injected with a lethal dose of sodium pentobarbital 24 h after the surgery, and two-mm thick brain tissue surrounding the path of the injection needle was collected and stored at $-80\,°C$ until use. All animal experiments conformed to the European Parliament Directive

(2010/63/EU) and were approved by the Institutional Animal Care and Use Committee at Harbin Medical University (No. HMUIRB-2008-06).

## Reverse transcription-quantitative polymerase chain reaction (RT-qPCR) analysis

Total RNA from the brain tissue of rats was extracted using Tri®-Reagent (Sigma, USA) according to the manufacturer's instructions. The quality and quantity of the RNA was detected using a NanoDrop One spectrophotometer (Thermo Fisher Scientific, Waltham, MA, USA). Reverse transcription of the extracted total RNA into cDNA was performed using the Transcriptor First Stand cDNA Synthesis Kit (Roche, Indianapolis, IN, USA). RT-qPCR was performed using the FastStart Universal SYBR® Green Master (ROX) (Roche, Penzberg, Germany) and quantified using the QuantStudio software (Thermo Fisher Scientific, Waltham, MA, USA) according the manufacturer's instructions. *GAPDH* was used as the endogenous reference gene. The sequence of all primers used in this study is provided in Table S1. The expression level of all mRNAs was determined using the $2^{-\Delta CT}$ method. A meaningful analysis between the two groups was performed by a paired $t$-test, and a $P$-value $< 0.05$ was considered statistically significant.

# RESULTS

## Identification of DEGs in ICH

The gene expression profile of the GSE24265 dataset contained data obtained from four perihematomal tissues and seven contralateral tissues (Table 1). The result of the normalization of data among the arrays of the series matrix files of the GSE24265 dataset is shown in Fig. S1. Using a $P$-value of $< 0.05$ and a [logFC] of $>2$ as the cut-off, a total of 67 DEGs, including 59 up-regulated and 8 down-regulated genes, were obtained on comparison of the expression profiles between the cerebral hemorrhage and control groups (Table 2). Volcano plots and heat maps for the distribution of DEGs were generated using R software (Figs. 1A and 1B).

## Functional annotation analysis of DEGs using GO

On GO analysis, the three most significant processes identified among the annotations of BP were inflammatory response, neutrophil chemotaxis, and immune response. In parallel, the three most significant processes identified among the annotations of MF were chemokine activity, CXCR chemokine receptor binding, and growth factor activity. Finally, the three most significant processes identified among the annotations of CC were extracellular space, extracellular region, and plasma membrane. According to the criteria, the results of all meaningful functional annotation analyses are summarized in Table 3 and Fig. 2.

## KEGG pathway analysis of DEGs

The results of the cell signaling pathway enrichment analysis of the DEGs identified a total of 29 meaningful pathways which were analyzed. The cellular signaling pathways clearly associated with brain damage included those for HIF-1, TNF, toll-like receptor, the NF-kappa B signaling pathway, cytokine-cytokine receptor interaction, and chemokine

**Table 1   Details for GEO ICH data.**

| Reference | Sample | GEO | Platform | PH | CW | CG |
|---|---|---|---|---|---|---|
| *Rosell et al. (2011)* | Brain | GSE24265 | GPL570 | 4 | 4 | 3 |

**Notes.**

PH,  perihematomal area;  CW,  contralateral white matter;  CG,  contralateral grey matter.

**Table 2   Screening DEGs in ICH by integrated microarray.**

| DEGs | Gene terms |
|---|---|
| Upregulated | CXCL8, CCL20, C15orf48, CXCL1, VMP1, AGPAT9, G0S2, CXCL3, PTX3, TFPI2, CYTIP, TNFAIP3, PPBP, HSPA6, BCL2A1, TREM1, CCL4, IL18RAP, PMAIP1, SERPINE1, HBD, TGFBI, PNP, SELE, TIMP1, IL6, NCF2, PLAUR, LAPTM5, TCIRG1, OLR1, HBB, DUSP5, ISG20,S100P, PROK2, METRNL, CTGF, CD93,HCAR3, ANXA2P2, C5AR1, TLR2, FGR,MARCO, HMOX1, TUBB6, CYB5R2 ADM, UPP1, SLC2A3, CD163, BCAT1,CLEC5A, CCL8, HLA-DRA, TNFRSF12A,ANXA2, S100A11. |
| Downregulated | NXPH1, PRRT2, SLITRK5, SOWAHA, CNTN3, CACNA2D3, MUM1L1, ZNRF3. |

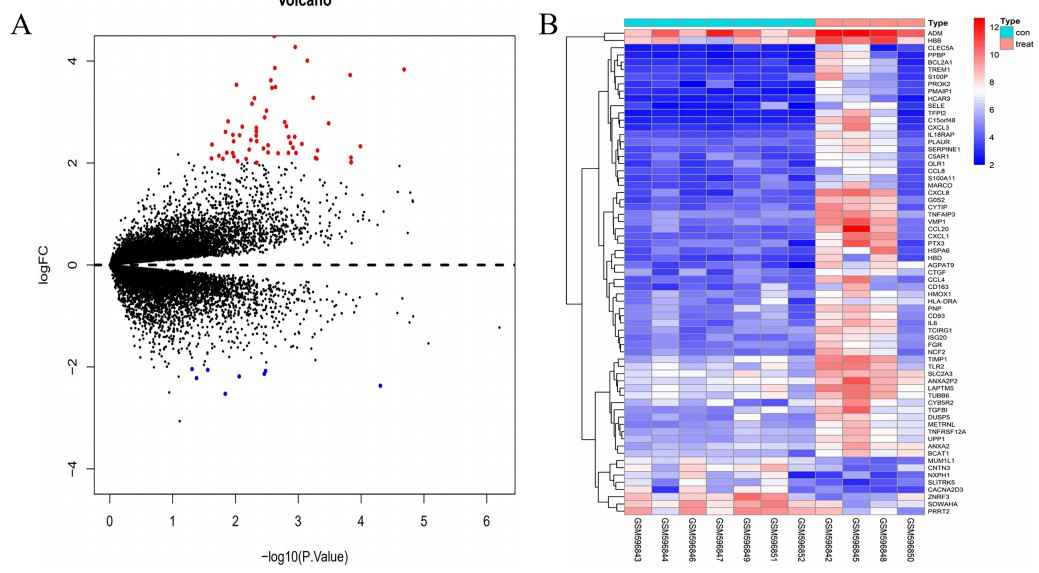

**Figure 1   Comparison of DEGs present in ICH and normal samples.** (A) The volcano plot for DEGs in GSE24265 data. *X*-axes index the -log (*P*-value) and *y*-axes index the log fold change. The red dots represent upregulated genes and the green dots represent downregulated genes. The black dots represent genes with no significant difference. FC is the fold change. (B) The expression data is represented as a data matrix wherein each row represents a gene and each column represents a sample. The blue coded bar above the heat map represents the normal sample set and the red coded bar represents the perihematomal sample. The expression level is described in terms of the color ratio of the upper left corner. Hierarchical clustering is shown by the top tree view, indicating the degree of relatedness in gene expression. Abbreviations: DEG, differentially expressed genes; ICH, intracerebral hemorrhage; FC, fold change.

**Table 3  Top 10 of the most significantly enriched GO terms.**

| Pathway ID | Terms | Gene count | *P*-value |
|---|---|---|---|
| BP | | | |
| GO:0006954 | Inflammatory response | 16 | 5.41E−12 |
| GO:0030593 | Neutrophil chemotaxis | 8 | 4.33E−09 |
| GO:0006955 | Immune response | 13 | 3.67E−08 |
| GO:0070098 | Chemokine-mediated signaling pathway | 7 | 2.35E−07 |
| GO:0006935 | Chemotaxis | 7 | 5.73E−06 |
| GO:0001525 | Angiogenesis | 8 | 1.75E−05 |
| GO:0032496 | Response to lipopolysaccharide | 7 | 3.10E−05 |
| GO:0090023 | Positive regulation of neutrophil chemotaxis | 4 | 7.02E−05 |
| GO:0071347 | Cellular response to interleukin-1 | 5 | 1.36E−04 |
| GO:0045765 | Regulation of angiogenesis | 4 | 2.00E−04 |
| CC | | | |
| GO:0005615 | Extracellular space | 19 | 5.27E−07 |
| GO:0005576 | Extracellular region | 19 | 6.79E−06 |
| GO:0005886 | Plasma membrane | 27 | 9.38E−04 |
| GO:0005604 | Basement membrane | 4 | 2.70E−03 |
| GO:0005887 | Integral component of plasma membrane | 13 | 3.43E−03 |
| GO:0070062 | Extracellular exosome | 19 | 6.75E−03 |
| GO:0009986 | Cell surface | 7 | 1.18E−02 |
| GO:0031093 | Platelet alpha granule lumen | 3 | 1.60E−02 |
| GO:0031012 | Extracellular matrix | 5 | 2.02E−02 |
| GO:0030666 | Endocytic vesicle membrane | 3 | 2.25E−02 |
| MF | | | |
| GO:0008009 | Chemokine activity | 7 | 2.25E−02 |
| GO:0045236 | CXCR chemokine receptor binding | 3 | 2.25E−02 |
| GO:0008083 | Growth factor activity | 5 | 2.25E−02 |
| GO:0002020 | Protease binding | 4 | 2.25E−02 |
| GO:0004872 | Receptor activity | 5 | 2.25E−02 |
| GO:0048306 | Calcpium-dependent protein binding | 3 | 2.25E−02 |
| GO:0008329 | Signaling pattern recognition receptor activity | 2 | 2.25E−02 |
| GO:0001849 | Complement component C1q binding | 2 | 2.25E−02 |
| GO:0005102 | Receptor binding | 5 | 2.25E−02 |
| GO:0004859 | Phospholipase inhibitor activity | 2 | 2.25E−02 |

Notes.
GO, Gene Ontology; BP, biological progress; CC, cellular component; MF, molecular function.

signaling pathways. The specific enriched pathways obtained from the analysis of the DEGs are summarized in Table 4 and Fig. 3.

## PPI network construction and hub gene identification

We uploaded 67 DEGs to the string online database to obtain the PPI network. A total of 57 DEGs were consequently extracted among the 67 DEGs uploaded when the interaction score was set to 0.15. The resulting PPI network contained a total of 57 nodes and 373 edges. Figure 4A presents the visualization of the network generated using Cytoscape software;
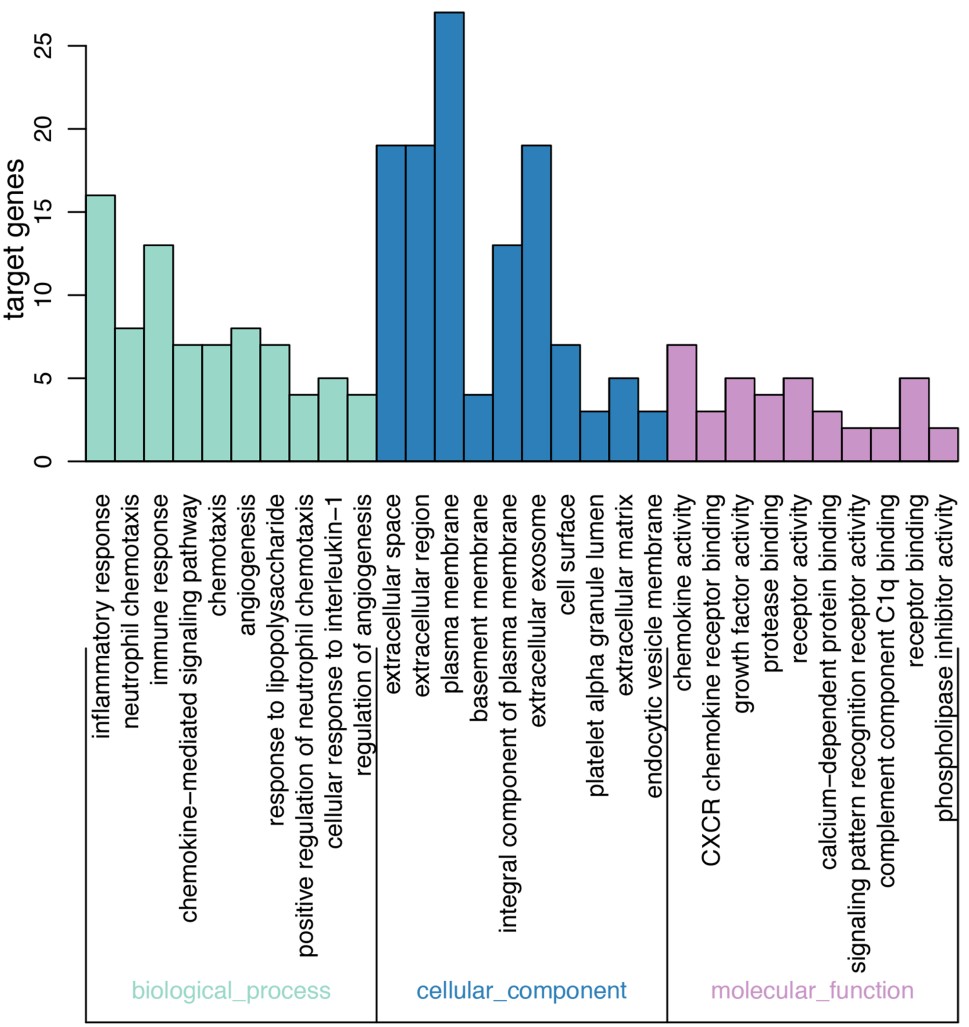

**Figure 2 Results of GO enrichment.** The abscissa represents the enriched GO, and the ordinate represents the number and ratio of the differentially expressed genes. Different colors represent different GO classes: Molecular function, Biological process, and Cellular component. Abbreviation: GO, gene ontology.

Figure 4B represents the ten selected hub genes with the highest degrees of connectivity that were selected as the hub genes. Prior reports indicate that the identified hub genes are closely related to ICH and brain damage. The subsequent GO function annotation analysis and KEGG pathway analysis results for hub genes are summarized in Tables 5 and 6.

## CMap analysis

We used CMap to analyze the previously selected DEGs and to identify small-molecule compounds with potential therapeutic application in ICH cases. Nine small-molecule compounds which exhibited high correlation with ICH are shown in Table 7. The three-dimensional chemical structures of the small-molecule compounds hecogenin, lidocaine,

**Table 4  Significantly enriched KEGG pathway.**

| Pathway ID | Terms | Gene count | *P*-Value | Genes |
|---|---|---|---|---|
| hsa04060 | Cytokine-cytokine receptor interaction | 10 | 1.59E−09 | *CCL4, CCL8, CXCL8, CCL20, TNFRSF12A, IL18RAP, CXCL3, PPBP, CXCL1, IL6* |
| hsa05323 | Rheumatoid arthritis | 7 | 3.50E−09 | *TCIRG1, CXCL8, CCL20, TLR2, CXCL1, HLA-DRA, IL6* |
| hsa05134 | Legionellosis | 6 | 9.62E−09 | *CXCL8, TLR2, CXCL3, CXCL1, IL6, HSPA6* |
| hsa04062 | Chemokine signaling pathway | 8 | 3.28E−08 | *CCL4, CCL8, CXCL8, CCL20, FGR, CXCL3, PPBP, CXCL1* |
| hsa04145 | Phagosome | 7 | 1.45E−07 | *TCIRG1, OLR1, NCF2, TLR2, TUBB6, HLA-DRA, MARCO* |
| hsa05144 | Malaria | 5 | 2.19E−07 | *IL6, CXCL8, HBB, SELE, TLR2* |
| hsa04668 | TNF signaling pathway | 6 | 5.37E−07 | *TNFAIP3, CCL20, CXCL3, SELE, CXCL1, IL6* |
| hsa05132 | Salmonella infection | 5 | 3.30E−06 | *CXCL1, CCL4, IL6, CXCL3, CXCL8* |
| hsa04621 | NOD-like receptor signaling pathway | 4 | 1.78E−05 | *CXCL1, IL6, TNFAIP3, UPP1* |
| hsa05321 | Inflammatory bowel disease (IBD) | 4 | 2.43E−05 | *HLA-DRA, IL6, IL18RAP, TLR2,* |
| hsa04064 | NF-kappa B signaling pathway | 4 | 1.04E−04 | *CCL4, TNFAIP3, BCL2A1, CXCL8* |
| hsa05143 | African trypanosomiasis | 3 | 1.09E−04 | *IL6, HBB, SELE* |
| hsa05146 | Amoebiasis | 4 | 1.36E−04 | *CXCL1, IL6, CXCL8, TLR2* |
| hsa04933 | AGE-RAGE signaling pathway in diabetic complications | 4 | 1.52E−04 | *IL6, SERPINE1, SELE, CXCL8* |
| hsa04066 | HIF-1 signaling pathway | 4 | 1.63E−04 | *IL6, SERPINE1, TIMP1, HMOX1* |
| hsa05142 | Chagas disease (American trypanosomiasis) | 4 | 1.69E−04 | *IL6, CXCL8, SERPINE1, TLR2* |
| hsa04620 | Toll-like receptor signaling pathway | 4 | 1.81E−04 | *CCL4, IL6, CXCL8, TLR2* |
| hsa05162 | Measles | 4 | 4.47E−04 | *IL6, TNFAIP3, HSPA6, TLR2* |
| hsa05120 | Epithelial cell signaling in Helicobacter pylori infection | 3 | 7.57E−04 | *CXCL1, TCIRG1, CXCL8* |
| hsa05140 | Leishmaniasis | 3 | 8.21E−04 | *HLA-DRA, NCF2, TLR2* |
| hsa05152 | Tuberculosis | 4 | 1.04E−03 | *TCIRG1, IL6, HLA-DRA, TLR2* |
| hsa05164 | Influenza A | 4 | 1.06E−03 | *HLA-DRA, IL6, HSPA6, CXCL8* |
| hsa04610 | Complement and coagulation cascades | 3 | 1.15E−03 | *C5AR1, PLAUR, SERPINE1* |
| hsa05169 | Epstein-Barr virus infection | 4 | 1.85E−03 | *HLA-DRA, FGR, TNFAIP3, HSPA6* |
| hsa05145 | Toxoplasmosis | 3 | 3.57E−03 | *HLA-DRA, HSPA6, TLR2* |
| hsa05332 | Graft-versus-host disease | 2 | 4.53E−03 | *HLA-DRA, IL6* |
| hsa04672 | Intestinal immune network for IgA production | 2 | 6.46E−03 | *HLA-DRA, IL6* |
| hsa05161 | Hepatitis B | 3 | 6.47E−03 | *IL6, CXCL8, TLR2* |
| hsa05150 | Staphylococcus aureus infection | 2 | 8.12E−03 | *HLA-DRA, C5AR1* |

**Notes.**
KEGG,  Kyoto Encyclopedia of Genes and Genomes.

and NU-1025, previously reported to be significantly associated with brain damage, are shown in Fig. 5.

## Corroboration of ten hub genes using RT-qPCR

RT-qPCR was performed with the total RNA extracted from the brain tissue of three pairs of rats used for the ICH model to confirm the expression levels of ten hub genes. According to the PPI network analysis, the ten core genes screened were *IL6, TLR2, CXCL1, TIMP1,*
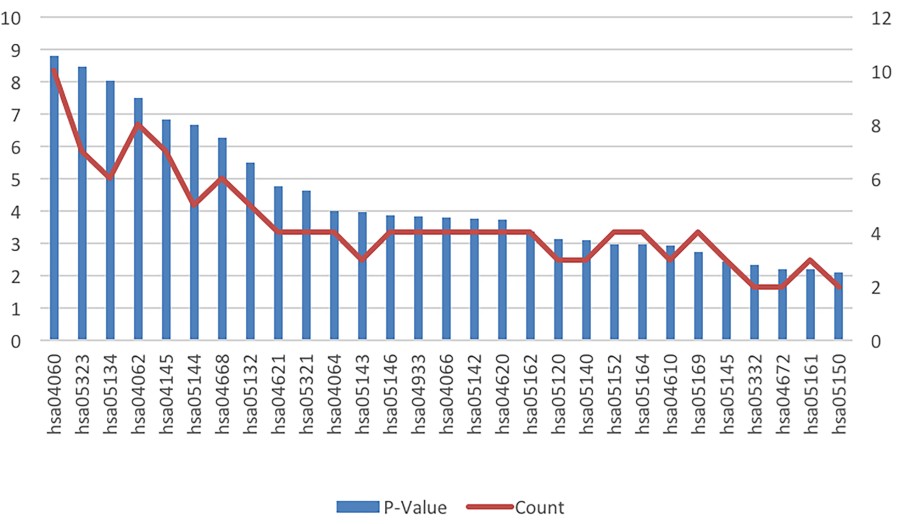

**Figure 3** **KEGG pathway analysis of the differentially expressed genes in ICH.** Abbreviations: KEGG, Kyoto Encyclopedia of Genes and Genomes; ICH, intracerebral hemorrhage.

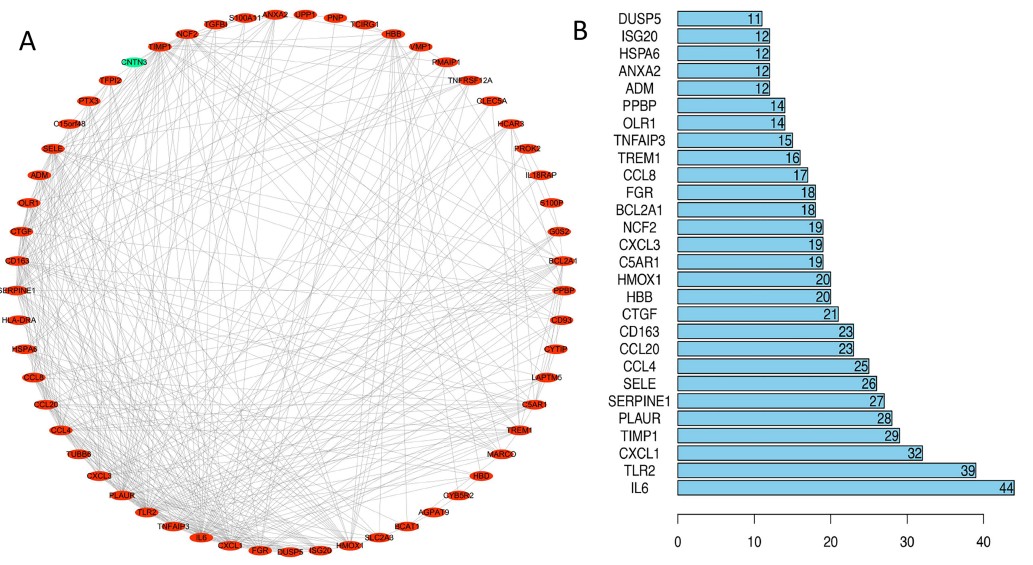

**Figure 4** **PPI network analysis.** (A) Using the STRING online database, a total of 57 DEGs (56 upregulated [red] and one downregulated gene [green]) were filtered into the DEG PPI network. (B) Of the DEGs with more than ten edges in the PPI network, the ten with the highest degrees of connectivity were selected as the hub genes; all are upregulated genes. Abbreviations: PPI, protein–protein interaction; DEGs, differentially expressed genes.

*PLAUR*, *SERPINE1*, *SELE*, *CCL4*, *CCL20*, and *CD163*. The expression levels of the ten hub genes were increased as part of the pathophysiological process of ICH. The results of the RT-qPCR experiment showed a significant increase in the expression of the hub genes (Fig. 6).

**Table 5  Significantly enriched GO of hub genes.**

| ID | Description | *P*-value | Count | Gene names |
|---|---|---|---|---|
| **BP** | | | | |
| GO:0009611 | Response to wounding | 4.58E−11 | 9 | CXCL1/ IL6/ CCL20/ SERPINE1/ TLR2/ SELE/ CCL4/ PLAUR/ CD163. |
| GO:0006954 | Inflammatory response | 1.45E−08 | 7 | CXCL1/ IL6/ CCL20/ TLR2/ SELE/ CCL4/ CD163. |
| GO:0006952 | Defense response | 6.44E−07 | 7 | CXCL1/ IL6/ CCL20/ TLR2/ SELE/ CCL4/ CD163. |
| GO:0042330 | Taxis | 2.27E−06 | 5 | CXCL1/ IL6/ CCL20/ CCL4/ PLAUR. |
| GO:0006935 | Chemotaxis | 2.27E−06 | 5 | CXCL1/ IL6/ CCL20/ CCL4/ PLAUR. |
| GO:0006955 | Immune response | 2.25E−04 | 4 | CXCL1/ IL6/ CCL20/ TLR2/ CCL4. |
| GO:0007596 | Blood coagulation | 6.88E−04 | 5 | IL6/ SERPINE1/ PLAUR. |
| GO:0050817 | Coagulation | 0.001396979 | 3 | IL6/ SERPINE1/ PLAUR. |
| GO:0007599 | Hemostasis | 0.001957914 | 3 | IL6/ SERPINE1/ PLAUR. |
| GO:0042742 | Defense response to bacterium | 0.001957914 | 3 | IL6/ CCL20/ TLR2. |
| GO:0006928 | Cell motion | 0.002191691 | 3 | IL6/ SELE/ CCL4/ PLAUR. |
| GO:0050878 | Regulation of body fluid levels | 0.002354575 | 3 | IL6/ SERPINE1/ PLAUR. |
| GO:0032101 | Regulation of response to external stimulus | 0.003084628 | 4 | IL6/ SERPINE1/ SELE. |
| GO:0042060 | Wound healing | 0.003701403 | 3 | IL6/ SERPINE1/ PLAUR. |
| GO:0032755 | Positive regulation of interleukin-6 production | 0.004681351 | 3 | IL6/ TLR2. |
| GO:0042127 | Regulation of cell proliferation | 0.006688038 | 3 | CXCL1/ IL6/ SERPINE1/ TIMP1. |
| GO:0042246 | Tissue regeneration | 0.012573365 | 2 | SERPINE1/ PLAUR. |
| GO:0032675 | Regulation of interleukin-6 production | 0.012647695 | 4 | IL6/ TLR2. |
| GO:0006953 | Acute-phase response | 0.02109503 | 2 | IL6/ CD163. |
| GO:0030162 | Regulation of proteolysis | 0.023703896 | 2 | PLAUR/ TIMP1. |
| GO:0050900 | Leukocyte migration | 0.02630658 | 2 | IL6/ SELE. |
| GO:0051091 | Positive regulation of transcription factor activity | 0.034722761 | 2 | IL6/ TLR2. |
| GO:0045765 | Regulation of angiogenesis | 0.037299313 | 2 | IL6/ SERPINE1. |
| GO:0031099 | Regeneration | 0.039227714 | 2 | SERPINE1/ PLAUR. |
| GO:0043388 | Positive regulation of DNA binding | 0.04115268 | 2 | IL6/ TLR2. |
| GO:0031349 | Positive regulation of defense response | 0.044992331 | 2 | IL6/ TLR2. |
| GO:0050727 | Regulation of inflammatory response | 0.049454644 | 2 | IL6/ SELE. |
| **CC** | | | | |
| GO:0009611 | Extracellular region | 2.86E−06 | 9 | CXCL1/ IL6/ CCL20/ SERPINE1/ SELE/ CCL4/ PLAUR/ CD163/ TIMP1. |
| GO:0006954 | Extracellular region part | 2.31E−04 | 6 | CXCL1/ IL6/ CCL20/ SELE/ CCL4/ TIMP1. |
| GO:0006952 | Extracellular space | 8.30E−04 | 5 | CXCL1/ IL6/ CCL20/ SELE/ CCL4. |

**Table 5** (*continued*)

| ID | Description | *P*-value | Count | Gene names |
|---|---|---|---|---|
| **MF** | | | | |
| GO:0005125 | Cytokine activity | 2.62E−04 | 4 | CXCL1/ IL6/ CCL20/ CCL4. |
| GO:0008009 | Chemokine activity | 4.35E−04 | 3 | CXCL1/ CCL20/ CCL4. |
| GO:0042379 | Chemokine receptor binding | 4.94E−04 | 3 | CXCL1/ CCL20/ CCL4. |
| GO:0019899 | Enzyme binding | 0.048323907 | 3 | SERPINE1/ SELE/ PLAUR. |

**Notes.**

GO, Gene Ontology; BP, biological progress; CC, cellular component; MF, molecular function.

**Table 6   Significantly enriched KEGG pathway of hub genes.**

| Pathway ID | Terms | Gene count | *P*-Value | Genes |
|---|---|---|---|---|
| hsa04668 | TNF signaling pathway | 4 | 5.54E−08 | *CXCL1, CCL20, IL6, SELE.* |
| hsa04066 | HIF-1 signaling pathway | 3 | 6.38E−06 | *IL6, SERPINE1, TIMP1.* |
| hsa04060 | Cytokine-cytokine receptor interaction | 3 | 9.63E−05 | *CXCL1, CCL20, IL6.* |
| hsa04621 | NOD-like receptor signaling pathway | 2 | 0.000199993 | *CXCL1,IL6.* |
| hsa04610 | Complement and coagulation cascades | 2 | 0.000364662 | *SERPINE1,PLAUR.* |
| hsa04620 | Toll-like receptor signaling pathway | 2 | 0.000671108 | *IL6,TLR2.* |
| hsa04062 | Chemokine signaling pathway | 2 | 0.00200085 | *CXCL1,CCL20.* |
| hsa04151 | PI3K-Akt signaling pathway | 2 | 0.006358451 | *IL6,TLR2.* |
| hsa04672 | Intestinal immune network for IgA production | 1 | 0.016724268 | *IL6* |
| hsa04623 | Cytosolic DNA-sensing pathway | 1 | 0.022711959 | *IL6* |
| hsa04115 | p53 signaling pathway | 1 | 0.02481751 | *SERPINE1* |
| hsa01521 | EGFR tyrosine kinase inhibitor resistance | 1 | 0.029714789 | *IL6* |
| hsa04068 | FoxO signaling pathway | 1 | 0.047713858 | *IL6* |

**Notes.**

KEGG, Kyoto Encyclopedia of Genes and Genomes.

**Table 7   Results of CMap analysis.**

| Rank | CMap name | Mean | *N* | Enrichment | *P*-value |
|---|---|---|---|---|---|
| 1 | Hecogenin | −0.613 | 4 | −0.931 | 0.00002 |
| 2 | Pralidoxime | −0.529 | 4 | −0.872 | 0.00056 |
| 3 | Metolazone | −0.282 | 5 | −0.779 | 0.00096 |
| 4 | Lidocaine | −0.258 | 5 | −0.766 | 0.0013 |
| 5 | NU-1025 | −0.579 | 2 | −0.956 | 0.00414 |
| 6 | Ambroxol | −0.269 | 4 | −0.774 | 0.00527 |
| 7 | Pipemidic acid | −0.336 | 3 | −0.846 | 0.00735 |
| 8 | Yohimbic acid | −0.345 | 3 | −0.845 | 0.00747 |
| 9 | Chlorphenesin | −0.354 | 4 | −0.74 | 0.00899 |

**Notes.**

CMap, Connectivity Map.

## DISCUSSION

The present study uses the GEO database to obtain gene expression profiles from patients with ICH and screens for DEGs. We perform functional enrichment analyses on the obtained DEGs to understand their biological functions. Meanwhile, we report meaningful

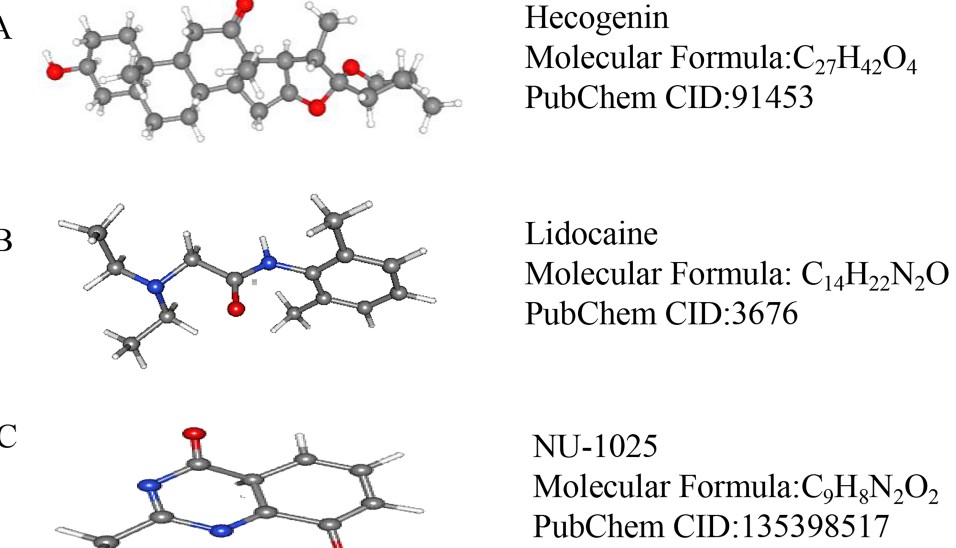

A  Hecogenin
Molecular Formula:$C_{27}H_{42}O_4$
PubChem CID:91453

B  Lidocaine
Molecular Formula: $C_{14}H_{22}N_2O$
PubChem CID:3676

C  NU-1025
Molecular Formula:$C_9H_8N_2O_2$
PubChem CID:135398517

**Figure 5** **The 3D conformers of the three compounds that may attenuate secondary injury after hematoma.** (A) Hecogenin, (B) Lidocaine, (C) NU-125. The 3D structures of the nine compounds were provided by PubChem (https://pubchem.ncbi.nlm.nih.gov/compound).

enrichment, by KEGG analysis, of pathways involved in triggering the regulatory pathway associated with secondary brain injury induced by ICH. Next, PPI analyses were carried out to identify the hub genes that may play a crucial regulatory role in the pathophysiology of ICH. At the same time, we established a rat model of ICH, and confirmed that the expression level of hub genes were consistent with the results of our analyses by RT-qPCR. In addition, several potential small-molecule compounds, related to ICH, were identified by CMap analysis. These may become effective drugs for treating ICH in future.

From the GSE24265 dataset, after comparing the gene expression data obtained from ICH tissue with that obtained from the contralateral brain tissues, we obtained 67 DEGs. GO analysis indicated that the main functional annotation results were consistent with those obtained by previous reports, i.e., a significant increase in inflammatory response promotes ICH-induced brain injury, which may be one of the main factors contributing to the poor prognosis in patients with ICH (*Boehme et al., 2018*; *Fu et al., 2018*). The immune response can antagonize the inflammatory response, induced after an episode of cerebral hemorrhage, to reduce the secondary damage to brain tissue, and thereby improve the recovery of nerve function (*Klebe et al., 2015*; *Lu et al., 2014*). Previous reports have indicated that the cellular signaling pathways identified in this study using a KEGG analysis were closely related to cerebral hemorrhage; those activated within a few hours after cerebral hemorrhage, including the TNF signaling pathway, which participates in the secondary inflammatory reactions and formation of cerebral edema around the hematoma (*Testai & Aiyagari, 2008*) and the HIF-1 signaling pathway, which contributes to the recovery of neural function after ICH by regulating the proliferation and differentiation of neural stem cells (*Yu et al., 2013*). The NF-kappa B signaling pathway, which is closely

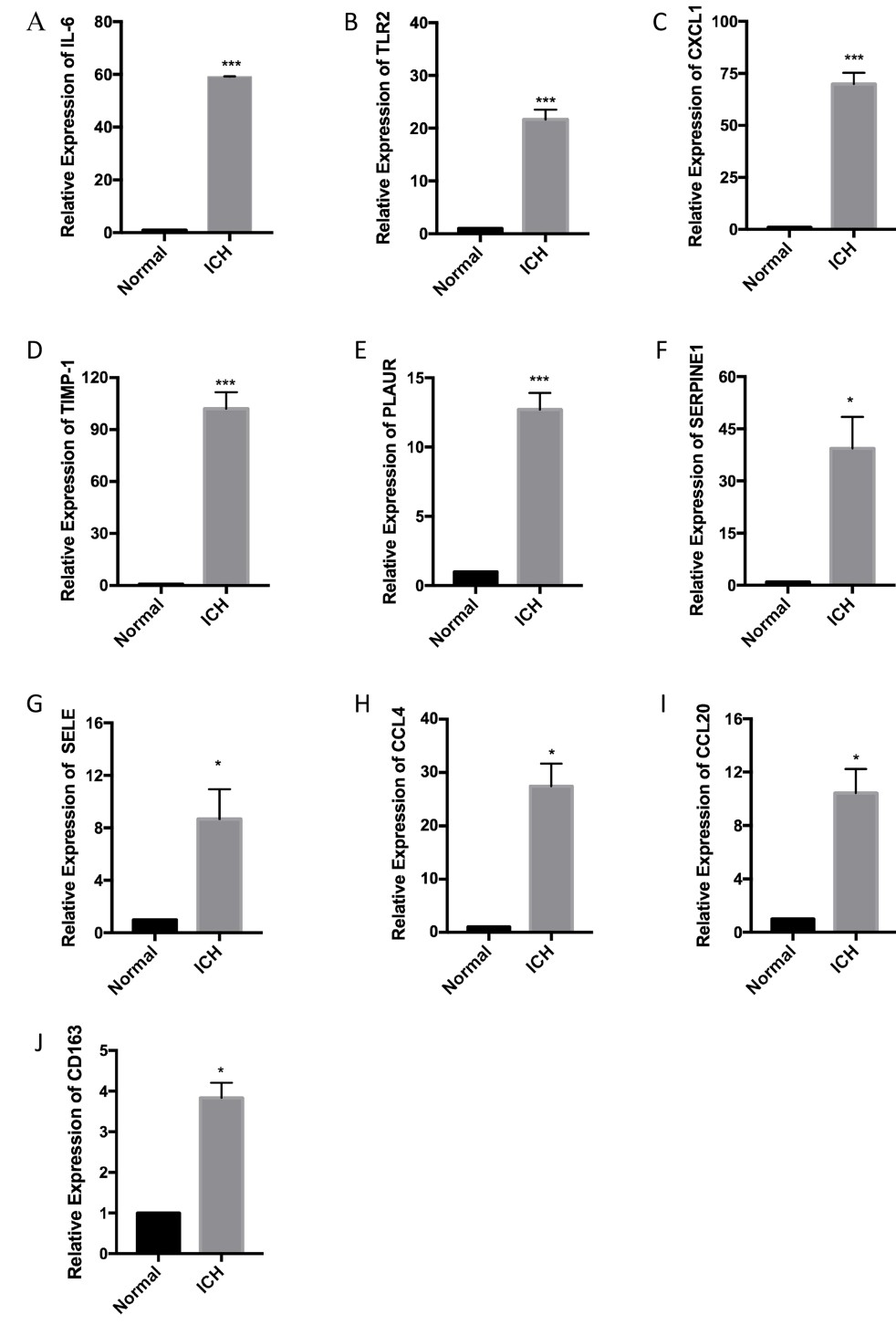

**Figure 6** Relative expression of the ten hub genes including (A) IL6, (B) TLR2, (C) CXCL1, (D) TIMP1, (E) PLAUR, (F) SERPINE1, (G) SELE, (H) CCL4, (I) CCL20, and (J) CD163 measured by RT-qPCR.

related to the death of cells surrounding the hematoma in the animal ICH model, and the activation of which is associated with deficiencies in neurological function (*Zhou et al., 2014*), was activated several minutes after the ICH event. The Toll-like receptor signaling pathway is significantly associated with a poor prognosis after ICH because it promotes the inflammatory response of ICH-induced brain injury (*Wang et al., 2014b*). Therefore, prior reports have been able to verify the accuracy of our results.

PPI networks are essential in most biological processes, play an important role in the development of many diseases and are very useful for the identification of drug targets, therefore demonstrating great potential for drug discovery (*Rabbani et al., 2018*). Compared to the original article which identified ICH related genes only by multiple gene expression analyses (*Rosell et al., 2011*), hub genes screened by constructing PPI networks are more relevant to the pathological processes of ICH and are more likely to be potential therapeutic targets. Therefore, we uploaded the DEGs to the STRING database to build a PPI network and further identify the hub genes that influence regulatory processes in the pathophysiology of ICH-induced brain injury. For instance, IL-6 was significantly elevated in the serum of patients with ICH, and its levels were clearly associated with Glasgow Coma Scale scores (*Dziedzic et al., 2002*). During brain injury, IL-6 signaling may not only induce the chemotactic migration of inflammatory cells to the site of lesion but may also promote T-cell polarization (*Chaudhry et al., 2017*). The inflammatory response plays a key role in ICH-induced brain injury, and hemoglobin causes toll-like receptors 2 and 4 to heterodimerize and aggravate brain damage (*Wang et al., 2014b*). Within 24 h of having a stroke, the expression level of CXCL1 in the cerebrospinal fluid of patients positively correlates with the ischemic area, indicating that CXCL1 may participate in the regulation of brain tissue damage (*Losy, Zaremba & Skrobanski, 2005*). The abnormal expression of TIMP1 can disrupt the integrity of the blood–brain barrier and this disruption may be involved in the regulation of the pathological process of ICH (*Wang et al., 2014a*). The urokinase-type plasminogen activator, encoded by the *PLAUR* gene, is involved in the development of the neural circuit of the cortex and in brain tissue remodeling after a brain injury (*Levi et al., 2001*). A few hours after the onset of ICH, because of red blood cell fragmentation, some products released by the lysed red blood cells around the site of the hematoma were involved in secondary brain damage. In this process, CD163 and hemoglobin form a complex which participates in the pathological process of secondary brain injury induced by ICH (*Lieber & Mocco, 2018*). *SERPINE1*, *CCL4*, and *CCL20* have not yet been reported in cases of IHC, but, on review of the literature, we found that these genes are associated with nerve injury, such as sterile inflammatory reaction and traumatic brain injury, which are pathologically similar to ICH, such as sterility inflammatory reaction (*Graber, Costine & Hickey, 2015*; *Kwiecien, 2018*; *Leonardo et al., 2012*). These hub genes, therefore, feature as potential prognostic or diagnostic markers, as well as potential therapeutic targets.

CMap is a database that covers the interactions among diseases, gene expression patterns, and small-molecule compounds. It is often used to screen for small-molecule compounds with therapeutic effects on some diseases (*Lamb et al., 2006*). Using CMap analysis, we identified nine small molecule drugs, two of which were lidocaine and NU-1025, both

previously reported to have a salutary effect in cases of brain injury. Lidocaine is commonly used as a local anesthetic and has been shown to have significant anti-inflammatory effects. A recent study demonstrated that lidocaine inhibited the release of inflammatory factors and alleviated microglial damage (*Jeong et al., 2012*). Moreover, lidocaine is considered to have a therapeutic effect in cases of nerve damage induced by cerebral hemorrhage. Recent studies have found that NU-1025, a poly (ADP-ribose) polymerase inhibitors, can attenuate neurological damage in animal models of ischemic stroke and traumatic brain injury; moreover, rats treated with NU1025 exhibited reduced infarction, significantly reduced edema volume, and significantly ameliorated neurological impairment (*Kaundal, Shah & Sharma, 2006*). Thus, NU-1025 can both mitigate the damage inflicted in the early stages of nerve injury and reduce the degree of subsequent neuroinflammation (*Curtin & Szabo, 2013*). Although the two aforementioned drugs have been reported to be associated with the treatment of ICH, the mechanisms underlying their therapeutic actions still require further research. In addition, we found that hecogenin has an anti-inflammatory effect that may contribute to the treatment of ICH (*Ingawale, Mandlik & Patel, 2019*; *Ingawale & Patel, 2016*; *Ingawale & Patel, 2018*).

However, this study had several limitations. First, the mechanisms of several hub genes in the pathological process of ICH remain unclear, warranting further study. Moreover, the effectiveness of our small molecule compound screening in reducing ICH-induced brain injury remains to be assessed.

## CONCLUSIONS

The hub genes identified by constructing a PPI network have potential in the development of new targets in the treatment of ICH and as prognostic markers in patients with ICH. The hub genes may be involved in the pathological process of ICH and ultimately affect their prognosis through TNF signaling pathway, HIF-1 signaling pathway, NF-kappa B signaling pathway, Toll-like receptor signaling pathway and Chemokine signaling pathway. In addition, this study identified small-molecule compounds that may be efficacious in the treatment of secondary brain injury induced by ICH.

### Funding
This work was supported by the China Postdoctoral Science Foundation (2017M620119); Heilongjiang Postdoctoral Fund (LBH-Z17108) and the Scientific Research Project of Heilongjiang Provincial Department of Health (Nos. 2013030). The funders had no role in study design, data collection and analysis, decision to publish, or preparation of the manuscript.

### Grant Disclosures
The following grant information was disclosed by the authors:
China Postdoctoral Science Foundation: 2017M620119.
Heilongjiang Postdoctoral Fund: LBH-Z17108.

Scientific Research Project of Heilongjiang Provincial Department of Health: 2013030.

## Competing Interests

The authors declare there are no competing interests.

## Author Contributions

- Zhendong Liu and Ruotian Zhang conceived and designed the experiments, performed the experiments, analyzed the data, contributed reagents/materials/analysis tools.
- Xin Chen contributed reagents/materials/analysis tools, authored or reviewed drafts of the paper.
- Penglei Yao and Jiawei Yao performed the experiments, prepared figures and/or tables.
- Tao Yan and Wenwu Liu performed the experiments, prepared figures and/or tables, performed the statistical analyses.
- Andrei Sokhatskii and Ilgiz Gareev analyzed the data, prepared figures and/or tables.
- Shiguang Zhao contributed reagents/materials/analysis tools, authored or reviewed drafts of the paper, approved the final draft.

## Animal Ethics

The following information was supplied relating to ethical approvals (i.e., approving body and any reference numbers):

All animal experiments conformed to the European Parliament Directive (2010/63/EU) and were approved by the Institutional Animal Care and Use Committee at Harbin Medical University (No. HMUIRB-2008-06).

## Data Availability

The data is available in the Gene Expression Omnibus: GSE24265.

## Supplemental Information

Supplemental information for this article can be found online at http://dx.doi.org/10.7717/peerj.7782#supplemental-information.

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
