# Peer review of "Identification of hub genes and small-molecule compounds related to intracerebral hemorrhage with bioinformatics analysis"

_PeerJ, doi:10.7717/peerj.7782_

## Round 0.1 · original submission · Major Revisions

Your manuscript has been reviewed by three experts in the field and their comments are shown below. As you will find, all of them raise substantial criticisms on it; one of them even recommends its rejection because of the lack of appropriate validation procedures. Considering these comments, I recommend the decision of major revision to the Editor-in-Chief. Please read the following comments carefully and revise the manuscript accordingly; if you don't think that some of these comments are appropriate to follow, state the reason clearly. i myself feel that one of the reviewers' comments that you should compare your results with those reported in original papers is particularly important.

[]

Reviewer 1 ·

Basic reporting

REVIEW For PeerJ #36749
May14, 2019
General Comments
I think this is an interesting bioinformatics study to identify hub genes associated with cerebral hemorrhage (ICH) as therapeutic targets and identify potential compounds for the treatment of ICH. The GSE24265 dataset was downloaded from the Gene Expression Omnibus (GEO) database to screen for differentially expressed genes (DEGs) for ICH. The function and pathway of DEGs were determined by gene ontology (GO) and Kyoto Encyclopedia of Genes and Genomes (KEGG) analyses. We constructed a protein-protein interaction (PPI) network to clarify the interrelationships of DEGs and select hub genes with significant interactions. Finally, the DEGs were analyzed by using the CMap tool to identify molecular compounds with potential therapeutic effects.
As one of the stroke physician who has been taking care of patients with ICH, authors results obtained from mainly data within the web site looks quite attractive, however, how and why authors determined the statistically significant criteria such as a P-value of < 0.05 and a [logFC] of > 2 as the cut-off condition for these datasets are not well described for general authors. If authors could revise these points, that would help a lot to initiate this type of bioinformatics stroke research very soon.

Experimental design

none

Validity of the findings

Authors reported new method and result of stroke research. It is quite interesting.

Additional comments

I think this is an interesting bioinformatics study to identify hub genes associated with cerebral hemorrhage (ICH) as therapeutic targets and identify potential compounds for the treatment of ICH. The GSE24265 dataset was downloaded from the Gene Expression Omnibus (GEO) database to screen for differentially expressed genes (DEGs) for ICH. The function and pathway of DEGs were determined by gene ontology (GO) and Kyoto Encyclopedia of Genes and Genomes (KEGG) analyses. We constructed a protein-protein interaction (PPI) network to clarify the interrelationships of DEGs and select hub genes with significant interactions. Finally, the DEGs were analyzed by using the CMap tool to identify molecular compounds with potential therapeutic effects.
As one of the stroke physician who has been taking care of patients with ICH, authors results obtained from mainly data within the web site looks quite attractive, however, how and why authors determined the statistically significant criteria such as a P-value of < 0.05 and a [logFC] of > 2 as the cut-off condition for these datasets are not well described for general authors. If authors could revise these points, that would help a lot to initiate this type of bioinformatics stroke research very soon.

·

Basic reporting

No comment

Experimental design

This manuscript presents interesting findings that are timely given the interest in post-ICH secondary injury, including inflammatory injury. However, the findings are somewhat limited in originality given the use of an existing gene expression dataset that has already been published, albeit using different analytical methodology.

While a prior publication from these publicly available data is cited, there is no discussion about the findings from this published manuscript or other potential manuscripts from these publicly available data. The authors should make more clear how their currently submitted work improves upon the previous publication.

Validity of the findings

See below

Additional comments

The manuscript seems to be missing any discussion of strengths and limitations.

There should be more discussion about the use of post-mortem brain tissue and the potential limitations for this methodology (e.g. what gene expression changes might occur in brain tissue in general just because it has been ischemic from death for up to 5 hours? Can any further information regarding this be provided?)

The substance that is being tested (e.g. brain tissue or peripheral blood) should be made more clear in the abstract

The statistical parameters for determining the differentially expressed genes should be made more clear in the abstract

The description of "we used online tools to perform GO enrichment and KEGG analyses" needs more explanation. What online tools? Subsequently, the description of DAVID and KOBAS is somewhat confusing, including the use of "its" following these terms. Does "its" apply to one of these tools or both?

For lines 123 and 168, the samples should not be described as "normal." They are actually contralateral brain samples from a patient with ICH, right? There may be changes even in this area just by the systemic effects of ICH, so it is not accurate to say that they are normal.

Statements that are made in the discussion should be more clear regarding what evidence is from ICH vs other disease processes, such as in line 202 to 203 when a number of papers are cited, and described as validating the currently submitted paper's findings, but none of those citations are from ICH studies.

Lines 224 to 227 are too general and seem inaccurate, including descriptions of "nerve damage" and " KEGG analysis is clearly involved in triggering the 226 regulatory pathway of brain injury."

Reviewer 3 ·

Basic reporting

Citation is inappropriate at more than one part, for example, line 77-78, "The gene expression profiles of ICH patients were obtained from the GEO database (http://www.ncbi.nlm.nih.gov / geo) (Guo et al. 2017), public database for researchers." but Guo2017 is not for GEO database. Also, line 88, Chen2016 is not for limma package (citation("limma") gives Ritchie2015).

Grammar; clumsy at several parts

Experimental design

Several unclear descriptions can be found. In line 87-88 " The data were normalized, and the DEGs were screened using the limma package in the R language (version 3.5.1) (Chen et al. 2016)." Who did normaliziation? If the authors did it, clarify the method.

Validity of the findings

Results seem to be fine in the context of applying publicly available tools.

Additional comments

Liu et al. downloaded expression data from GEO database and applied downloadable publicly available analysis software. The general concern is the lack of replication/validation effort for the analysis result.

Also, most of the molecules listed in Table 5 are quite unlikely to contribute to ICH treatment, which might be due to the lack of replication.

Iff the author collects their own biological materials for statistical validation, or try biological confirmation of the findings, such work would be considered for publication.

---

## Round 0.2 · Major Revisions

Your revised MS has been reviewed by the same three referees. One of them now recommends its acceptance while the others still raise several points. Please read their comments carefully and re-revise your MS accordingly.

Reviewer 1 ·

Basic reporting

REVIEW For PeerJ #36749R1
July 24, 2019
General comments;
In this version of manuscript, authors revised the text. Now authors have appropriately responded to my questions and further upgraded the paper quality for easy understanding.

Experimental design

I think this paper has been appropriately designed.

Validity of the findings

None.

Additional comments

REVIEW For PeerJ #36749R1
July 24, 2019
General comments;
In this version of manuscript, authors revised the text. Now authors have appropriately responded to my questions and further upgraded the paper quality for easy understanding.

·

Basic reporting

See comments

Experimental design

See comments

Validity of the findings

See comments

Additional comments

I continue to have the following concerns:



The authors did not address strengths and limitations of the study as I recommended in the first set of reviewer comments.



The manuscript is very confusing in that terms are used such as "gene mutations" and genetic abberations" and yet this is a study of microarray analysis for GENE EXPRESSION, not GWAS or DNA sequencing for DNA mutations. The terminology used needs to be more accurate and consistent throughout the manuscript.



The conclusion of the abstract is too general for the reader to actually take away conclusions related to the main results.



The authors add the rat model to the methods and results as separate sections, but there is no integration of those animal model findings with the human data in the abstract or the discussion or conclusion of the manuscript.


I do not understand this statement in the discussion: "The remaining analytical results are not described, and the accuracy of our results can be further verified by the prior reports."



The authors did not actually include information (i.e. main findings) from other manuscript(s) published using this publicly available dataset (i.e. Rosell et al), as I recommended in the first reviewer comment.



The authors should include that the timing from ICH onset to sample collection was within 4 days. They should also include that there were 2 women and 2 men and the ages. This information is easily found on the NCBI website about GSE24265.

Reviewer 3 ·

Basic reporting

Re-submitted manuscript must have tracks of the changes made in the manuscript.

The grammar has been improved.

Experimental design

I'm not an expert in this aspect (mouse experiment)

Validity of the findings

This reviewer appreciates the efforts of the authors to validate their bioinformatics results.

Additional comments

In the original manuscript, the authors selected 67 DEGs with P < 0.05 and logFC > 2 without any validation of this analysis result. Note that the family-wise error rate of this analysis is 1-(1-0.05)^M where M would be ~20000 genes or so. So this is only regarded as a narrative data, without any statistical inference. Then the authors did GO or pathway enrichment analysis, constructed PPI network, or performed CMap analysis using these 67 genes. Similarly, these are narrative data without scientific validity (unless other types of statistical evidence, such as Bayesian, are stated).

In this re-submission, the authors did validation analysis using mouse model for 10 PPI-hub genes and successfully validated that these are ICH-related DEGs.

Now this reviewer is largely satisfied with their claim that these 10 genes are ICH-related DEGs. Then, GO, pathway and CMap analyses should be done with these 10 genes, not the non-validated 67 DEGs as a whole.

---

## Round 0.3 · accepted · Accept

I personally checked your re-revised manuscript and confirm that the points are reasonably addressed. Thus, I have made the decision of its acceptance. Thanks for your patience.